# Structural congenital anomalies in resource limited setting, 2023: A systematic review and meta-analysis

Yohannes Fikadu Geda[1]*, Yirgalem Yosef Lamiso[1‡], Tamirat Melis Berhe[1‡], Seid Jemal Mohammed[1‡], Samuel Ejeta Chibsa[2‡], Tadesse Sahle Adeba[1‡], Kenzudin Assfa Mossa[1‡], Seblework Abeje[1‡], Molalegn Mesele Gesese[3‡]

1 Wolkite University, Wolkite, Ethiopia, 2 Mettu University, Mettu, Ethiopia, 3 Wolaita Sodo University, Wolaita Sodo, Ethiopia

‡ YYL, TMB, SJM, SEC, TSA, KAM, SA and MMG are coauthors on this work.
* nechsar@gmail.com

## Abstract

### Background

A number of studies have looked at neonatal structural birth defects. However, there is no study with a comprehensive review of structural anomalies. Therefor we aimed to verify the best available articles to pool possible risk factors of structural congenital anomalies in resource limited settings.

### Setting

Genuine search of the research articles was done via PubMed, Scopes, Cochrane library, the Web of Science; free Google database search engines, Google Scholar, and Science Direct databases. Published and unpublished articles were searched and screened for inclusion in the final analysis and Studies without sound methodologies, and review and meta-analysis were not included in this analysis.

### Participants

This review analyzed data from 95,755 women who have birthed from as reported by primary studies. Ten articles were included in this systematic review and meta-analysis. Articles which have no full information important for the analysis and case reports were excluded from the study.

### Results

The overall pooled effect estimate of structural congenital anomalies was 5.50 [4.88–6.12]. In this systematic review and meta-analysis maternal illness effect estimate (EI) with odds ratio (OR) = 4.93 (95%CI 1.02–8.85), unidentified drug use OR = 2.83 (95%CI 1.19–4.46), birth weight OR = 4.20 (95%CI 2.12–6.28), chewing chat OR = 3.73 (95%CI 1.20–6.30), chemical exposure OR = 4.27 (95%CI 1.19–8.44) and taking folic acid tablet during pregnancy OR = 6.01 (95%CI 2.87–14.89) were statistically significant in this meta-regression.

**Data Availability Statement:** The supporting data for these findings is accessible and is included with the paper as a supplementary file.

**Funding:** The author) received no specific funding for this work.

**Competing interests:** The authors have declared that no competing interests exist.

**Abbreviations:** CA, Congenital Anomalies; CI, Confidence Interval; EI, Effect Estimate; NOS, Newcastle-Ottawa Scale; OR, Odds Ratio; PRISM, Preferred Reporting Items of Systematic Reviews and Meta-Analysis; WHO, World Health Organization.

## Conclusions

The overall pooled effect estimate of structural congenital anomalies in a resource limited setting was high compared to better resource countries. On the Meta-regression maternal illness, unidentified drug use, birth weight, chewing chat, chemical exposure and never using folic acid were found to be statistically significant variables Preconception care and adequate intake of folic acid before and during early pregnancy should be advised.

## Background

Congenital abnormalities (CA), often known as birth defects, are prenatal structural or functional changes that can be identified during pregnancy, labor, and delivery, or even years after birth [1, 2]. We may classify it as primary or minor abnormalities based on the magnitude of the structural and functional conditions and the need for medical support or treatment [1, 3, 4].

Congenital malformation can damage a variety of organs, depending on the stage of development at the time when the harm occurred [5, 6]. The most common congenital anomalies, according to some research, are those of the central nervous system [5, 6]. Heart and neural tube deformities and Down syndrome are the most frequent congenital abnormalities [1, 7].

About 50% of birth defects do not have a definite cause; however, some genetic problems, environmental agents and infectious agents are known risk factors [8, 9]. The contribution of parental chromosomal abnormalities is about 2–4%; the contributions of anatomical abnormality, endocrine factors, and antiphospholipid antibody syndrome are about 10%–15%, 20%–27%, and 17–20%, respectively [10]. Many of the known causes of congenital abnormalities can be prevented through vaccination and appropriate prenatal care during pregnancy [8, 11, 12].

Overall, it is estimated that around 7.9 million (6%) children were born with congenital abnormalities [1, 2]. According to a World Health Organization (WHO) report, congenital abnormalities account for between 17% and 42% of infant mortality [13], from 2000 to 2016, approximately 295,000 children died in the first 28 days following birth [14].

Congenital abnormalities were the fifth leading cause of death in children under five years of age, accounting for over 10% of all under-five deaths [15]. Birth defects are estimated at 94 per cent [16] and 96% of deaths due to congenital anomalies occur in low and middle-income countries (LMIC) [15].

In sub-Saharan Africa, birth defects are thought to be responsible for 10% of deaths of children under the age of five [15]. Between 2.8% and 15.9% of people in Nigeria are said to have congenital anomalies [5], and it was 0.9–17.3% in Ethiopia [17–19].

Some of the congenital anomalies that have been reported in Ethiopia include anencephaly, hydrocephalus, spina bifida, meningomyelocele, umbilical hernia, upper and lower limb, cardiovascular system, digestive system, abdominal wall, unspecified congenital malformations, Down syndrome, genitourinary system, head, face, and neck defects, cleft lip and palate, clubfoot, and hernias [17–21].

Maternal age, the percentage of women who live in cities, educational attainment, nutritional status, usage of herbal and over-the-counter drugs, supplementation with folic acid, alcohol intake, and employment status are socio-demographic factors are associated with congenital abnormalities [22–24]. The likelihood of a successful pregnancy is typically close to 80% if the causes of birth defects are found and treated [25]. If not, congenital anomalies can

have lifelong effects and can be treated with both surgical and non-surgical methods [20]. Despite this, congenital anomalies have received little attention in low resource settings, leaving a significant knowledge and understanding gap regarding their prevalence and risk factors [15, 23, 24].

Despite the fact that several primary articles have been written about potential risk factors for structural congenital defects in settings with limited resources, there is no study that might be used as benchmark with pooled value in such settings. Therefor this systematic review and meta-analysis was carried out to examine the pooled potential risk factors of structural congenital abnormalities in resource-limited settings. The result and conclusion of this study will provide scientific information for program planners, other researchers, and policy developers to improve service delivery. Besides, it will be useful for health professionals in using evidence based practices to provide the services.

## Methods

### 2.1 Study design and setting

The authors assessed the PROSPERO database (https://www.crd.york.ac.uk/PROSPERO/) for all published or ongoing research available related to the title to skip any further duplication. Accordingly the result brought that there were no ongoing or published articles in the area of this title. Therefor this review and meta-analysis was registered in the PROSPERO database with an identification number of CRD42022384838 on 28/12/2022. This review and meta-analysis was conducted to verify the pooled possible risk factors of structural congenital anomalies in resource limited settings. Scientific consistency was formulated by using PRISMA checklist [26].

### 2.2 Information source

A systematic and genuine search of the research articles was done via the following listed databases. PubMed, Scopes, Cochrane library, the Web of Science, free Google database search engines, Google Scholar, and Science Direct search engines were included in the review. We have used the keywords and MeSH terms (S1 File).

The search was performed using the following key search terms: "AND" and "OR" boolean operators individually and in combination with each other. Moreover, the reference lists of all the included studies were also searched to identify any other studies that may have been missed by the search strategy. The search for all research was done from October 10[th] to December 5[th], 2022 without limiting the publication dates of the literatures.

### 2.3 Eligibility criteria

**2.3.1 Inclusion criteria.** Published articles in national and international journals, and unpublished articles from institutional repositories conducted in resource limited settings with a result of possible risk factors of structural congenital anomalies were included in this study. Published and unpublished articles were searched and screened for inclusion in the final analysis. This study included in available observational study designs (cross-sectional studies and case-control studies). All research that was published, master's thesis found in institutional repositories, and PhD dissertation accessed from the repositories till the final date of data analysis and submission of this manuscript to this journal were included in accordance with these criteria.

During the beginning of our search 42 studies were found of which 13 were skipped due to duplication and the rest 29 studies were identified for eligibility. From 29 studies 10 were

excluded from highlight review on their abstracts and 19 studies assessed for full text of this 9 studies excluded because of not relevant to the current review and the remaining 10 studies were included in the final meta-analysis of this study (Fig 1).

**2.3.2 Exclusion criteria.** Studies without sound methodologies, and review and meta-analysis were not included in this analysis. Those articles which have no full information important for the analysis and case reports were excluded from the study. Duplication of results in studies and outcome variable measures with inconsistency were excluded from the final analysis. Studies, which incorporate other types of congenital anomalies were excluded (Fig 1).

## 2.4 Operational definition

**Structural congenital anomalies**: are structural changes, whether substantial or slight, that have a significant impact on an individual's health or appearance and often demand for medical attention.

**Resource-limited setting**: were categorized as low-income nations by the World Bank, a global alliance of nations devoted to eradicating poverty, which determined that they had the weakest economy [27].

**Unidentified drug use**: using a drug that has not been approved for the client by the clinician and that might affect the mother's or fetus natural physiological function.

**Birth weight**: Birth weight less than 2,500 grams was considered as low birth weight, whereas birth weight exceeding this was seen as normal [28].

**Chewing chat**: In this study "chewing chat" was marked with "yes" if a mother of a newborn with at least a weekly chat chewing experience.

**Chemical exposure**: A mother of the neonate who has jobs exposing to chemicals in a measurement of the amount and the frequency with which, a substance comes into contact.

Never using folic acid: A person whose most recent pregnancy was preceded by no folic acid use.

## 2.5 Quality assessment and data extraction

The basic quality of included research articles was evaluated using the Newcastle-Ottawa Scale (NOS). NOS were designed to assess the quality of observational research articles in meta-analyses. Data from this study were extracted by the two authors (YFG and YYL) using a standardized checklist for extracting data on an Excel sheet.

This meta-analysis uses the PRISMA flowchart to differentiate and select items of significance to the analysis. Initially, duplicate types of studies were not included using the Endnote version X8.1 referencing tool. Articles were excluded by adding highlights by going through their titles and abstracts before evaluating the entire text. Full-text studies or research results have been evaluated for other studies. Based on the aforementioned eligibility criteria; items have been assessed for eligibility.

Data were extracted using the standardized data extraction tool in considering the name of the first author, publication year, country of study, author's affiliation, sample size, magnitude of antenatal exercise and their 95% confidence interval (Table 1). All literacies were independently verified by the two authors (YFG and YYL). Where disagreements have occurred, the articles have been reviewed by one of the authors (TMB) and used as final mediation and admissibility decision. To obtain the pooled possible risk factor of congenital anomalies random effect model was used with a p value less than 0.001.

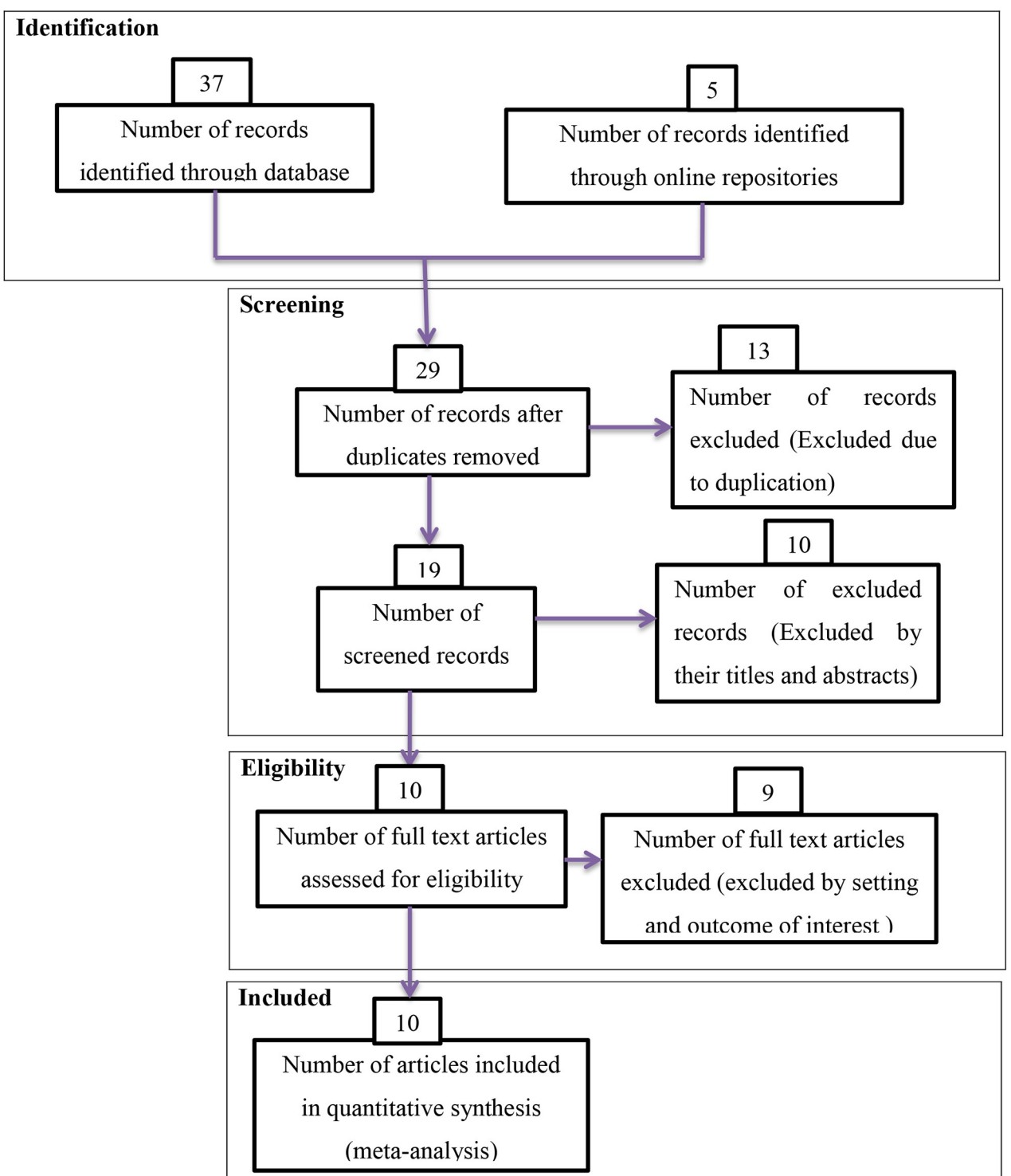

**Fig 1. PRISMA flow diagrams of included studies in the Systematic Review and Meta-analysis on possible risk factors of congenital anomalies in resource limited setting, 2022.**

**Table 1. Descriptive summary of included articles to pool possible risk factors of congenital anomalies in low resource setting, 2022.**

| Authors | Year | Design | Study area | Sample size | Number of cases | inclusion criteria of cases |
|---|---|---|---|---|---|---|
| Abebe et al. [39] | 2021 | Case-control | Southwestern Ethiopia | 1,138 | 251 | Live birth or fresh stillbirth |
| Bekalu et al. [36] | 2019 | Cross-sectional | Jimma | 754 | 31 | Total births with CAs |
| Eshete et al. [38] | 2020 | Case-control | Addis Ababa | 116 | 3215 | Total births with CAs |
| Feredegn et al. [33] | 2018 | Cross-sectional | Addis Ababa | 271 | 97 | Live births |
| Gedamu et al. [31] | 2021 | Cross-sectional | Bishoftu | 2,218 | 23 | Live births |
| Jemal et al. [37] | 2021 | Case-control | Arsi | 418 | 105 | Total births externally visible defects |
| Mekonnen et al. [30] | 2021 | Cross-sectional | Bahir Dar | 11,177 | 69 | Total births with CAs |
| Musa et al. [34] | 2020 | Cross-sectional | Addis Ababa | 116 | 71 | Live births |
| Silesh et al. [35] | 2021 | Cross-sectional | Jimma | 3,346 | 199 | Live births |
| Taye et al. [32] | 2019 | Cross-sectional | Addis Ababa and Amhara | 76,201 | 1518 | Live births |

## 2.6 Data synthesis and analysis

Both systematic review and meta-analysis were and the software used for the analysis was STATA version 14.0. Quantitative reviews were conducted to determine the overall pooled possible risk factors of structural congenital anomalies in low resource setting. The degree of heterogeneity between the included studies was evaluated by determining the p-values of $I^2$-test statistics. $I^2$ test statistics scores of 0, 25, 50, and 75% were taken as no, low, moderate, and high degrees of heterogeneity, respectively [29]. Due to the observed high heterogeneity across studies, we used a random effect model to assess pooled estimate. Publication bias was checked by funnel plot. A p-value of less than 0.05 was used as the cutoff point for statistical significance of publication bias.

## 2.7 Ethics approval and consent to participate

Ethical approval for this study was not applicable since this study was analyzed from secondary data without patient identification.

## Results

### 3.1 Characterization of included studies

Ten articles were included in this systematic review and meta-analysis and it was summarized in Table 1. Seven articles of the included study had used cross-sectional study design [30–36] while three articles were case control studies [37–39] with a sample size ranging from 418 in Arsi [37] to 76,201 in Addis Ababa and Amhara region [32].

In relation to the geographical location in which the study was conducted, six articles were from central Ethiopia [31–34, 37, 38], one study from Northern Ethiopia [30] and three studies from south western Ethiopia [35, 36, 39] (Table 1).

### 3.2 Publication bias

Bias among the included studies was checked by the funnel plot at a 5% significance level. The funnel plot was symmetry, and showed no statistical significance for the presence of publication bias for each variable. Egger test was done and verified that there was no small-study effects with P = 0.063 (Fig 2).

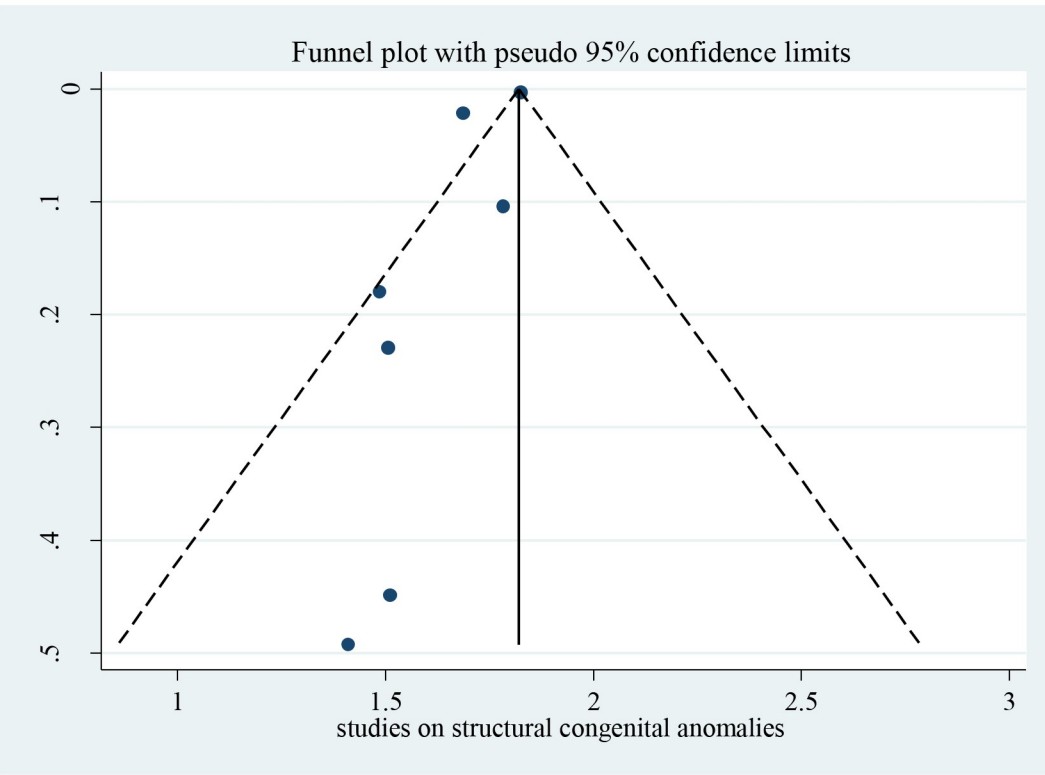

**Fig 2. Funnel plot for studies on possible risk factors of congenital anomalies in resource limited setting, 2022.**

### 3.3 Structural congenital anomalies

Only cross-sectional studies eligible for the analysis have reported prevalence of structural congenital anomalies. The overall pooled effect estimate of structural congenital anomalies was 5.50 with 95% confidence interval of 4.88 to 6.12 (Fig 3).

### 3.4 Possible risk factors of Congenital anomalies in low resource setting

In this systematic review and meta-analysis previous history of abortion, maternal illness, history of alcohol intake during pregnancy, unidentified drug use, birth weight, chewing chat, chemical exposure and taking folic acid tablets during pregnancy were statistically significant at one or more of the included primary studies. However, maternal illness, unidentified drug use, birth weight, chewing chat, chemical exposure and taking folic acid tablets during pregnancy were staying statistically significant in this meta-regression.

This review analyzed data from 95,755 women who have birth to estimate the pooled possible risk factors of congenital anomalies in low resource setting. A total of 10 (9 published and one unpublished) articles was included in this review (Table 1).

**3.4.1 Maternal illness.** Meta-analysis pooling of aggregate data using the random-effects and inverse-variance model with Der-Simonian-Laird estimate of $tau^2$ was done for 'maternal illness' separately. Test of the pooled overall effect provides 4.93 with a 95%CI 1.02–8.85; which shows neonates of woman with previous illness were 4.93 times more likely to have structural congenital anomalies compared to women who have no history of illness (Table 2).

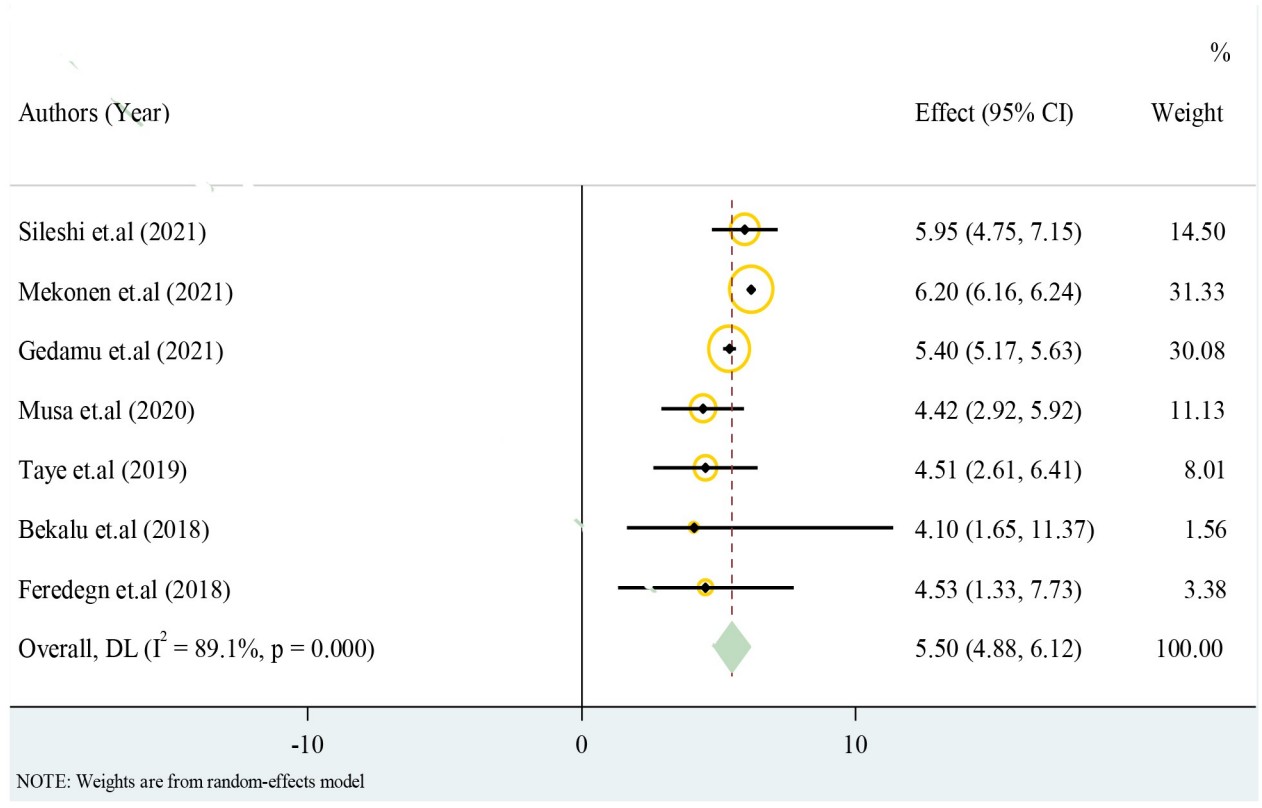

**Fig 3. Forest plot for structural congenital anomalies in resource limited setting, 2022.**

**3.4.2 Unidentified drug use.** Meta-regression of 'unidentified drug use' with the data using the random-effects and inverse-variance model shows unidentified drug use during pregnancy had significantly associated with congenital anomalies in low resource setting. Women who had a history of unidentified drug use during pregnancy were 2.83 times more likely to have structural congenital anomalies compared to women who haven't history of drug use during pregnancy (Table 2).

**3.4.3 Birth weight.** Birth weight was found to be statistically significant variable associated with structural congenital anomalies in resource limited settings. Neonates with birth weight less than 2.5kg were more likely to have structural congenital anomalies compared to neonates with birth weight greater than or equal to 2.5kg. The meta-regression of birth weight considering random-effects and inverse-variance model had 4.2 overall effects with a 95% confidence interval of 2.12 to 6.29 (Table 2).

**3.4.4 Chat chewing.** Pregnant women who have chat chewing experience were found to be significantly having structural congenital anomalies in the primary studies. The overall pooled effect women chewing chat were 3.73 times more likely to have structural congenital anomalies compared to women who never chew chat (Table 2).

**3.4.5 Never use folic acid.** Never using folic acid was a statistically significant variable in number of primary studies and in the meta-regression as well. Pregnant women who haven't used iron folate were 6.01 times more likely to have neonates with structural congenital anomalies compared to who have used folic acid during and before pregnancy (Table 2).

**3.4.6 Subgroup analysis to pool possible risk factors of structural congenital anomalies.** The listed individual variables were repeated in the analysis of a study within subgroups

**Table 2. Meta-regression result of pooled possible risk factors of congenital anomalies in resource limited setting, 2022.**

| Authors | Effect | [95% Conf. Interval] | % Weight |
|---|---|---|---|
| **Maternal illness** | | | |
| Jemal et al. [37] | 6.10 | 2.39–15.57 | 35.23 |
| Bekalu et al. [36] | 4.30 | 1.65–11.37 | 64.77 |
| Overall pooled | 4.93 | 1.02–8.84 | 100.00 |
| **Unidentified drug use** | | | |
| Abebe et al. [39] | 3.4 | 2.0–5.8 | 42.77 |
| Feredegn et al. [33] | 2.2 | 1.1–4.0 | 56.35 |
| Bekalu et al. [36] | 15.1 | 5.5–40.2 | 0.88 |
| Overall pooled | 2.83 | 1.19–4.46 | 100 |
| **Birth weight <2.5kg** | | | |
| Mekonnen et al. [30] | 4.56 | 2.76–7.55 | 75.55 |
| Gedamu et al. [31] | 3.10 | 1.23–9.65 | 24.45 |
| Overall pooled | 4.20 | 2.12–6.28 | 100.00 |
| **Chat chewing** | | | |
| Bekalu et al. [36] | 3.41 | 1.50–7.90 | 62.88 |
| Jemal et al. [37] | 4.76 | 1.57–14.47 | 15.48 |
| Abebe et al. [39] | 3.93 | 1.30–12.20 | 21.64 |
| Overall pooled | 3.73 | 1.20–6.30 | 100.00 |
| **Chemical exposure** | | | |
| Jemal et al. [37] | 4.76 | 1.57–14.47 | 41.70 |
| Abebe et al. [39] | 3.93 | 1.26–12.17 | 58.30 |
| Overall pooled | 4.27 | 1.19–8.44 | 100.00 |
| **Never use folic acid** | | | |
| Jemal et al. [37] | 0.57 | 0.41–0.73 | 25.00 |
| Abebe et al. [39] | 1.78 | 1.38–2.17 | 24.99 |
| Bekalu et al. [36] | 4.10 | 3.89–4.22 | 25.00 |
| Gedamu et al. [31] | 17.64 | 17.50–17.78 | 25.00 |
| Overall pooled | 6.01 | 2.87–14.89 | 100.00 |

of subjects defined by a subgrouping variable. Each variable was presented with $I^2$ and P-value to see the heterogeneities between studies (Fig 4).

## Discussion

The overall pooled effect estimate of structural congenital anomalies in resource limited setting was 5.50 with 95% confidence interval of 4.88 to 6.12. On the Meta-regression maternal illness, unidentified drug use, birth weight, chewing chat chemical exposure and never using folic acid were found to be statistically significant variables which might be the possible risk factors of congenital anomalies in low resource setting.

This study had come up with maternal illness was one of the possible risk factors of congenital anomalies in resource limited settings. Consistently a study concluded that maternal exposure to illness, fever, and medication (particularly aspirin) may increase the risk of congenital anomalies [40]. Another study conducted on the association between congenital anomalies and gestational diabetes mellitus stated that there was an increased rate of congenital anomalies in offspring of women with diabetes [41]. A study reported that first trimester maternal influenza exposure was associated with an increased risk of any congenital anomaly [42]. This

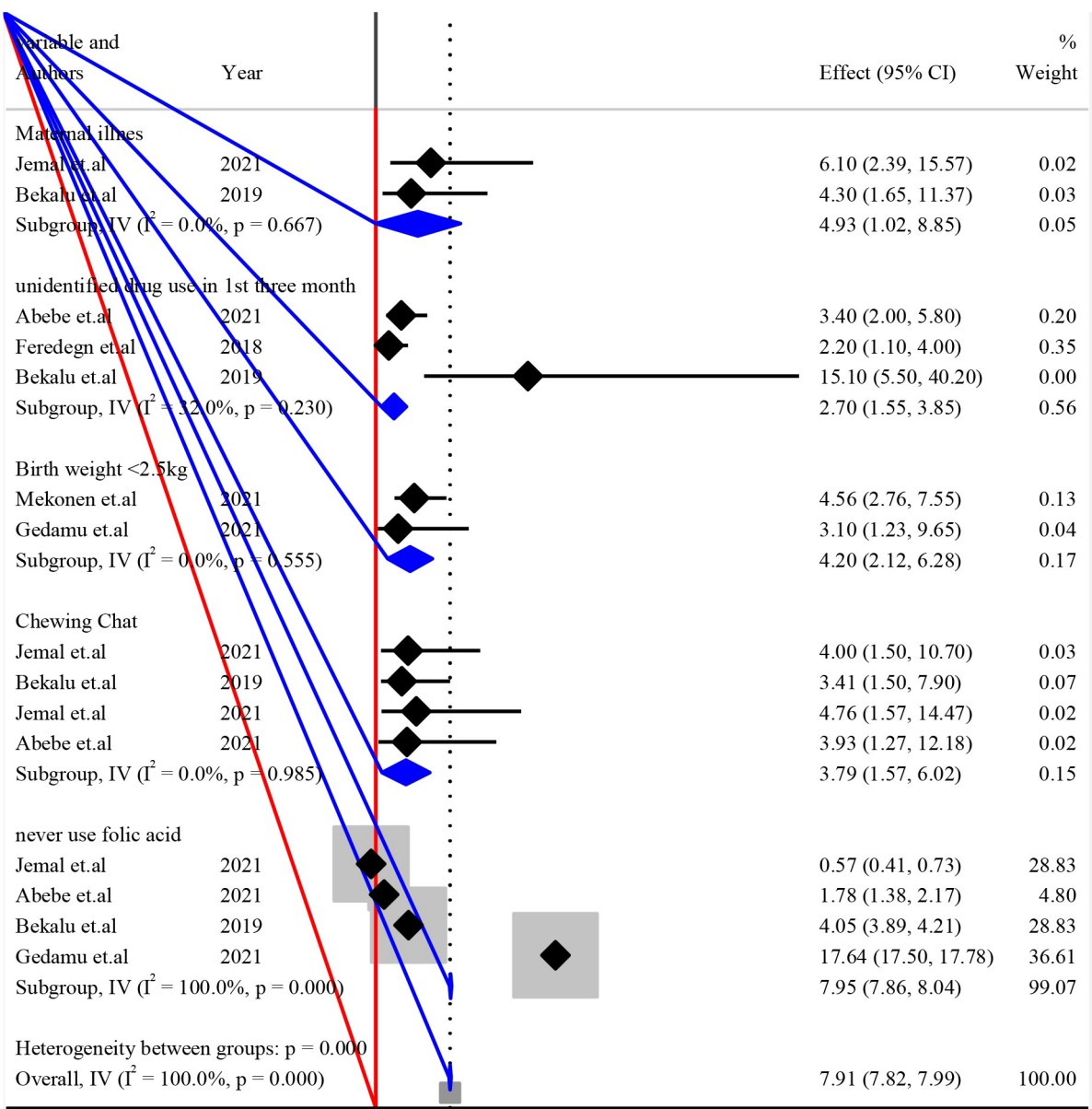

**Fig 4. Forest plot of subgroup analysis by variables for pooled possible risk factors of congenital anomalies in resource limited setting, 2022.**

might be due to causative agent of such diseases could pass the placental barrier and cause structural anomalies. However, experimental studies need to be conducted to confirm the associations.

This study verified that the unidentified drug use was one of the possible risk factors of congenital anomalies in resource limited settings. Similarly, studies showed that first-trimester paroxetine, fluoxetine, sertraline and anti-thyroid drug therapy exposures were associated with a significant increase in the risk of major congenital anomalies [43–48].

This might be due to those drugs need to be categorized as drugs demonstrated fetal abnormalities. However, positive evidence of fetal risk in human exists, but the benefit from use in

pregnant women may be acceptable in spite of the risk. Such as a life threatening situation for which a safer agent cannot be used.

In this study birth weight was found to be significantly affected congenital anomalies in low resource setting. In the same manner studies reported that congenital anomaly increased the risk of in-hospital mortality and was associated with short-term neonatal morbidities in low birth weight infants [49, 50]. Another study states that the prevalence of neonates with low birth weight and congenital anomalies was very high [51]. This might be due to fetuses with structural anomalies may have difficulties in using nutrients provided by the placenta appropriately due to their deformation. Moreover fetuses with structural anomalies are more likely to have functional anomalies which might disrupt metabolism and growth in the uterus.

In this study, chewing chat was the possible risk factor of congenital anomalies in low resource setting. Similarly a study conducted in Yemen clarifies that women who had chewed chat were more likely to have a poor neonatal outcome [52]. This might show that chemical in chat could pass the placental barrier and cause the anomalies. In the other hand, consumption of chat affects the growth of the fetus by inhibiting blood flow from the uterus to the placenta, which in turn affects the normal growth of the fetus.

This study revealed that chemical exposure was the possible risk factor of congenital anomalies in low resource setting. Consistently a study investigated of the strong association between congenital anomalies and mothers' exposure to air pollution by nitrogen dioxide during pregnancy by combining risk estimates for a variety of air pollutants [53]. Another study reported that evidence for an effect of ambient air pollutants on congenital anomaly risk [54]. This might show that chemicals in work or living environment pregnant women could cause structural congenital anomalies. This may suggest that specially work environment of pregnant women needs to be screened for potential chemicals able to cause anomalies.

This study showed that clients who had never taken folic acid tablet were more likely to develop congenital anomalies, or clients who had taken folic acid tablet were less likely to develop congenital anomalies compared to those who haven't taken folic acid. Similarly a study states that maternal preconception folic acid supplementation was significantly associated with the risk of congenital anomalies [55]. A study shows a robust estimate of the positive association between maternal folate supplementation and a decreased risk of congenital anomalies [56]. This might be due to intake of folic acid prior to conception and during the early stages of pregnancy plays an important role in preventing structural congenital anomalies.

## Strength and limitation

This systematic review and meta-analysis brings summative analysis of all primary studies conducted in resource limited settings. All variables available in each article were assessed for significance in the pooled effect. Pooled possible risk factors of structural congenital anomalies were obtained and Pooled significant variables were identified. But this systematic review might not be generalized to all countries with the resource limited settings. Because the available primary studies were conducted in some of the regions of low income countries.

## Conclusion and recommendation

The overall pooled effect estimate of structural congenital anomalies in a resource limited settings was high compared to those countries with better resources. On the Meta-regression maternal illness, unidentified drug use, birth weight, chewing chat, chemical exposure and never using folic acid were found to be statistically significant variables which might be the possible risk factors of congenital anomalies in low resource setting.

Therefore Women with illnesses like diabetes mellitus should be advised to have preconception care and antenatal care contact by the health offices in all resource limited settings.

Prevention based on reproduction options includes teratogen information like chewing chat, providing drugs without checking their teratogenicity, chemical exposure and prenatal screening for fetal anomalies should be done by all hospitals delivering preconception and pregnancy services.

## Supporting information

**S1 Table. PRISMA checklist.**
(DOCX)

**S2 Table. NOS quality assessment.**
(DOCX)

**S1 File. MeSH terms.**
(DOCX)

**S1 Dataset.**
(RAR)

## Acknowledgments

Authors of the primary research used on this systematic review and meta-analysis never need to be missed from acknowledgment.

## Author Contributions

**Conceptualization:** Yohannes Fikadu Geda.

**Data curation:** Yohannes Fikadu Geda, Yirgalem Yosef Lamiso, Tamirat Melis Berhe, Seid Jemal Mohammed, Samuel Ejeta Chibsa, Tadesse Sahle Adeba, Kenzudin Assfa Mossa, Seblework Abeje, Molalegn Mesele Gesese.

**Formal analysis:** Yohannes Fikadu Geda, Yirgalem Yosef Lamiso, Tamirat Melis Berhe, Seid Jemal Mohammed, Samuel Ejeta Chibsa, Tadesse Sahle Adeba, Kenzudin Assfa Mossa, Seblework Abeje, Molalegn Mesele Gesese.

**Funding acquisition:** Yohannes Fikadu Geda, Yirgalem Yosef Lamiso, Tamirat Melis Berhe, Seid Jemal Mohammed, Samuel Ejeta Chibsa, Tadesse Sahle Adeba, Kenzudin Assfa Mossa, Seblework Abeje, Molalegn Mesele Gesese.

**Investigation:** Yohannes Fikadu Geda, Yirgalem Yosef Lamiso, Tamirat Melis Berhe, Seid Jemal Mohammed, Samuel Ejeta Chibsa, Tadesse Sahle Adeba, Kenzudin Assfa Mossa, Seblework Abeje, Molalegn Mesele Gesese.

**Methodology:** Yohannes Fikadu Geda, Yirgalem Yosef Lamiso, Tamirat Melis Berhe, Seid Jemal Mohammed, Samuel Ejeta Chibsa, Tadesse Sahle Adeba, Kenzudin Assfa Mossa, Seblework Abeje, Molalegn Mesele Gesese.

**Project administration:** Yohannes Fikadu Geda, Yirgalem Yosef Lamiso, Tamirat Melis Berhe, Seid Jemal Mohammed, Samuel Ejeta Chibsa, Tadesse Sahle Adeba, Kenzudin Assfa Mossa, Seblework Abeje, Molalegn Mesele Gesese.

**Resources:** Yohannes Fikadu Geda, Yirgalem Yosef Lamiso, Tamirat Melis Berhe, Seid Jemal Mohammed, Samuel Ejeta Chibsa, Tadesse Sahle Adeba, Kenzudin Assfa Mossa, Seblework Abeje, Molalegn Mesele Gesese.

**Software:** Yohannes Fikadu Geda, Yirgalem Yosef Lamiso, Tamirat Melis Berhe, Seid Jemal Mohammed, Samuel Ejeta Chibsa, Tadesse Sahle Adeba, Kenzudin Assfa Mossa, Seblework Abeje, Molalegn Mesele Gesese.

**Supervision:** Yohannes Fikadu Geda, Yirgalem Yosef Lamiso, Tamirat Melis Berhe, Seid Jemal Mohammed, Samuel Ejeta Chibsa, Tadesse Sahle Adeba, Kenzudin Assfa Mossa, Seblework Abeje, Molalegn Mesele Gesese.

**Validation:** Yohannes Fikadu Geda, Yirgalem Yosef Lamiso, Tamirat Melis Berhe, Seid Jemal Mohammed, Samuel Ejeta Chibsa, Tadesse Sahle Adeba, Kenzudin Assfa Mossa, Seblework Abeje, Molalegn Mesele Gesese.

**Visualization:** Yohannes Fikadu Geda, Yirgalem Yosef Lamiso, Tamirat Melis Berhe, Seid Jemal Mohammed, Samuel Ejeta Chibsa, Tadesse Sahle Adeba, Kenzudin Assfa Mossa, Seblework Abeje, Molalegn Mesele Gesese.

**Writing – original draft:** Yohannes Fikadu Geda.

**Writing – review & editing:** Yohannes Fikadu Geda, Yirgalem Yosef Lamiso, Tamirat Melis Berhe, Seid Jemal Mohammed, Samuel Ejeta Chibsa, Tadesse Sahle Adeba, Kenzudin Assfa Mossa, Seblework Abeje, Molalegn Mesele Gesese.

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
