## [Decision Letter · Decision Letter 0]

31 May 2023

PONE-D-23-05213Structural congenital anomalies in resource limited setting, 2023: A systematic review and meta-analysisPLOS ONE

Dear Dr. Yohannes Fikadu Geda,

Thank you for submitting your manuscript to PLOS ONE. After careful consideration, we feel that it has merit but does not fully meet PLOS ONE’s publication criteria as it currently stands. Therefore, we invite you to submit a revised version of the manuscript that addresses the points raised during the review process. Please submit your revised manuscript by Jul 15 2023 11:59PM. If you will need more time than this to complete your revisions, please reply to this message or contact the journal office at plosone@plos.org. Please include the following items when submitting your revised manuscript:A rebuttal letter that responds to each point raised by the academic editor and reviewer(s). You should upload this letter as a separate file labeled 'Response to Reviewers'.A marked-up copy of your manuscript that highlights changes made to the original version. You should upload this as a separate file labeled 'Revised Manuscript with Track Changes'.An unmarked version of your revised paper without tracked changes. You should upload this as a separate file labeled 'Manuscript'.

We look forward to receiving your revised manuscript.

Kind regards,

Abay Woday Tadesse

Academic Editor

PLOS ONE

Journal Requirements:

- https://doi.org/10.1016/S0095-4543(21)00887-3

- https://doi.org/10.1016/j.heliyon.2022.e11657

In your revision ensure you cite all your sources (including your own works), and quote or rephrase any duplicated text outside the methods section. Further consideration is dependent on these concerns being addressed.

Additional Editor Comments:

Title: Structural congenital anomalies in resource limited setting, 2023: A systematic review and meta-analysis

General: The authors tried to review the title which is an important public health problems. The authors should include the following comments, suggestions, and questions. In addition, the manuscript could be benefited from native English language editors to improve the language of the manuscript to improve it, otherwise it is suitable for publication with current shape.

Background:

Authors should clearly indicate why they conduct this review and meta-analysis?

Methods:

Please revise your "MeSH terms" in PubMed and attach your search strategy for each databases you have listed as an additional file (supplementary file).

The eligibility criteria is not clearly stated. Where is PICO?

Who did searching, data extraction and what type study quality check method was employed in your case?

Results:

Figure 1 describe those excluded by setting and outcomes in your ‘study characteristics’ part and indicate the difference clearly. Similarly, indicate studies which were removed due to quality if any.

Table 1 indicated all studies were performed in Ethiopia, so why you talking about resource limited setting? The title is misleading the readers.

How taking folic acid tablet during pregnancy OR =6.01 (95%CI 2.87-14.89) was positively associated with outcome in contrast to what already known?

you missed some important papers:

doi: 10.1177/2333794X20974218

Additional files required for:

searching techniques for all databases

quality assessment grading using Newcastle-Ottawa Scale

Discussion:

Generally, the discussion is very shallow and needs revision.

The first and second paragraph is not related to discussion. The author should have focused on the summary finding in the first paragraph.

Authors should link their finding to previous studies and provide the possible reason behind for the complement and disagreement of the findings.

What are the implications of your findings?

Recommendations should summarize in one or two sentences based on your conclusion.

Did your findings support preconception care and vaccination care that you recommended?

Reviewers' comments:

Reviewer's Responses to Questions

**Comments to the Author**

1. Is the manuscript technically sound, and do the data support the conclusions?

Reviewer #1: Yes

2. Has the statistical analysis been performed appropriately and rigorously? 

Reviewer #1: I Don't Know

3. Have the authors made all data underlying the findings in their manuscript fully available?

Reviewer #1: No

4. Is the manuscript presented in an intelligible fashion and written in standard English?

Reviewer #1: No

5. Review Comments to the Author

Reviewer #1: The authors tried to review the title which is an important public health problems. The neural tube defect is common problems in low and middle income countries due to different factors. The underutilization of health services including preconception service in this regions makes things complicated. Having the overall understanding of the problem is very relevant. However, authors should incorporate or answer the following comments, suggestions, and questions and improve the language of the manuscript to improve it, otherwise it is suitable for publication with current shape.

Title: Structural congenital anomalies in resource limited setting, 2023: A systematic review and meta-analysis

Is resource limited setting is well known in World Bank economic classification of countries? I.e., high income countries, middle income countries, low income countries etc.

Abstract

Your background claim that there is a paucity of studies with a comprehensive review of structural anomalies; hence what is this gaps has to do with pooling the existing evidences?

Make your aim in line with your title.

Who did searching, data extraction and what type study quality check method was employed in your case?

How taking folic acid tablet during pregnancy OR =6.01 (95%CI 2.87-14.89) was positively associated with outcome in contrast to what already known?

The authors should indicate how outcomes variables and explanatory variables such as maternal illness, birth weight, chewing chat, chemical exposure, taking folic acid during pregnancy, unidentified drugs etc. were measured.

Did your study/review support preconception care and vaccination care that you recommended?

Background

Indicate the location of the evidences and its context, i.e., whether it is in the globe, LMICs, SSA etc. and also indicate type of evidence whether it is review, report, article etc.

“……, there is no study that serves as a standard for such settings.” So are you going to conduct primary study?

Authors failed to indicate why they are performing this systematic review and meta-analysis. Why large geographic coverage was considered despite huge heterogeneity that it could result in?

How authors pooling observational studies identified risk factors?

Authors should have indicated how stakeholders benefitted from systematic review and meta-analysis, whether researchers, policy or program designers, clinicians, etc.

Method

Information source did not indicate the MeSH terms correctly, for instance the MeSH term for congenital abnormalities is ‘Congenital Abnormalities’ not “Congenital Abnormalities/abnormalities".

The sample search of PubMed is not well organized. The key terms are not exhaustive.

The eligibility is not well explained point by points. For instance how you managed studies that reported proportion of outcome only, non-structural congenital anomalies, etc.

Where is PICO or other?

The description following preceding figure 1 is study characteristic that should have been incorporated to result section.

Where are study selection and data extraction subtitles?

“Egger test was done and verified that there was no small-study effects.” Don’t you think this is part of result?

Result

Figure 1 describe those excluded by setting and outcomes in your ‘study characteristics’ part and indicate the difference clearly. Similarly, indicate studies which were removed due to quality if any.

Table 1 indicated all studies were performed in Ethiopia, hence why you talking about resource limited setting? Why not you say simple ‘Ethiopia’. The title mislead readers.

The funnel plot indicate asymmetry, not symmetry; what is cutoff point for egger test to declare insignificance? Please indicate with proper citation in method section.

Discussion

The first and second paragraph is not related to discussion. The author should have focused on the summary finding in the first paragraph.

“Selection and characterization of included studies” what selection have to do in the result section?

“To obtain the pooled possible risk factor of congenital anomalies random effect model was used with a p value less than 0.001.” Don’t you think this should have included in the method section?

How facility-based studies reported prevalence of structural congenital anomalies?

“Possible risk factors of Congenital anomalies in low resource setting” how authors identified risk factors using cross-sectional studies?

Authors should link their finding to previous studies and provide the possible reason behind for the complement and disagreement of the findings. For instance how maternal illness related to tubal defect and what is the reason behind? How chat chewing related to tubal defect? Have you included studies from eastern part of the country where chat chewing is relatively high? What is the pathophysiology behind? Why https://www.ncbi.nlm.nih.gov/pmc/articles/PMC7672758/ not included? Which specific maternal illness is associated with tubal defects? How? Why?

What are the implications of your findings?

Recommendations are not usually listed as you did, instead summarize in one or two sentences based on your conclusion.

Additional files

Where is searching techniques for all databases?

Where is quality assessment grading using Newcastle-Ottawa Scale?

6. PLOS authors have the option to publish the peer review history of their article (what does this mean?). If published, this will include your full peer review and any attached files.

Reviewer #1: **Yes: **Kasiye Shiferaw (PhD)

---

## [Author Response · Author response to Decision Letter 0]

16 Jun 2023

Dear editors we appreciate you considering publishing our work in one of your prestigious journal. Editors' and reviewers' issues have been thoroughly addressed, and adjustments have been made as necessary.

Dear reviewer, we are excited to incorporate your insightful criticism to improve our article. As a result, we changed the manuscript; the changes are indicated in the revised version by track changes. Below are detailed responses to the reviewer concerns.

Response: Revised was made in accordance with PLOSONE formatting samples.

- https://doi.org/10.1016/S0095-4543(21)00887-3

- https://doi.org/10.1016/j.heliyon.2022.e11657

In your revision ensure you cite all your sources (including your own works), and quote or rephrase any duplicated text outside the methods section. Further consideration is dependent on these concerns being addressed.

Response: The manuscript was revised based on the advice provided and altered as necessary. YFG is the corresponding author of this document and the second study listed with doi above, and both studies' designs are meta-analyses. Even with considerable cutting and paraphrasing, this might result in some text overlap on the methods and materials section.

Response: Amendments were made based on the recommendations. 

Response: Justification was made on the revised manuscript. 

Response: Corrected 

Response: Captions for supporting files are included in the revised version of the manuscript 

Reviewer #1: The authors tried to review the title which is an important public health problems. The neural tube defect is common problems in low and middle income countries due to different factors. The underutilization of health services including preconception service in this regions makes things complicated. Having the overall understanding of the problem is very relevant. However, authors should incorporate or answer the following comments, suggestions, and questions and improve the language of the manuscript to improve it, otherwise it is suitable for publication with current shape.

Title: Structural congenital anomalies in resource limited setting, 2023: A systematic review and meta-analysis

Is resource limited setting is well known in World Bank economic classification of countries? I.e., high income countries, middle income countries, low income countries etc.

Response: Your concern was considered significant, and the operational definition of a resource-limited setting was added to the methods and materials section in the amended paper.

Abstract

Your background claim that there is a paucity of studies with a comprehensive review of structural anomalies; hence what is this gaps has to do with pooling the existing evidences?

Make your aim in line with your title.

Response: The error has been fixed in the current version. It was meant to imply that there are no studies that include a comprehensive review.

Who did searching, data extraction and what type study quality check method was employed in your case?

Response: The initials of authors conducted searching and data extraction were described in quality assessment and data extraction subsection of the Methods and materials section. The basic quality of included research articles was evaluated using the Newcastle-Ottawa Scale and included as a supporting file in the current version. 

How taking folic acid tablet during pregnancy OR =6.01 (95%CI 2.87-14.89) was positively associated with outcome in contrast to what already known?

Response: This study claims that never using folic acid was positively associated with outcome. 

The authors should indicate how outcomes variables and explanatory variables such as maternal illness, birth weight, chewing chat, chemical exposure, taking folic acid during pregnancy, unidentified drugs etc. were measured.

Response: Those variables are described in operational definition subsection of methods and materials section. 

Did your study/review support preconception care and vaccination care that you recommended?

Response: Vaccination was removed from the recommendation section. All statistically significant factors in this study that could result in congenital abnormalities could be avoided if the client received preconception care. That’s why preconception care included in the recommendation. 

Background

Indicate the location of the evidences and its context, i.e., whether it is in the globe, LMICs, SSA etc. and also indicate type of evidence whether it is review, report, article etc.

Response: The background was changed to reflect this. 

“……, there is no study that serves as a standard for such settings.” So are you going to conduct primary study?

Response: Last paragraph of the background was intensively revised. 

Authors failed to indicate why they are performing this systematic review and meta-analysis. Why large geographic coverage was considered despite huge heterogeneity that it could result in?

Response: Even though congenital abnormalities could be significantly reduced with preconception care, providing preconception care has been hampered in all resources-limited settings which make them homogeneous. 

How authors pooling observational studies identified risk factors?

Response: The term "possible risk factors" was chosen for this study because, despite the fact that the primary studies were observational studies, it more accurately characterizes the components listed here than the term "associated factors."

Authors should have indicated how stakeholders benefitted from systematic review and meta-analysis, whether researchers, policy or program designers, clinicians, etc.

Response: It’s indicated in the final paragraph of the background section. 

Method

Information source did not indicate the MeSH terms correctly, for instance the MeSH term for congenital abnormalities is ‘Congenital Abnormalities’ not “Congenital Abnormalities/abnormalities".

The sample search of PubMed is not well organized. The key terms are not exhaustive.

The eligibility is not well explained point by points. For instance how you managed studies that reported proportion of outcome only, non-structural congenital anomalies, etc.

Response: Revised and provided as a supporting file. 

Where is PICO or other?

Response: Included in table 1

Where are study selection and data extraction subtitles?

Response: It’s included in quality assessment and data extraction subsection

“Egger test was done and verified that there was no small-study effects.” Don’t you think this is part of result?

Response: Yes it’s, and corrected accordingly. 

Result

Figure 1 describe those excluded by setting and outcomes in your ‘study characteristics’ part and indicate the difference clearly. Similarly, indicate studies which were removed due to quality if any.

Table 1 indicated all studies were performed in Ethiopia, hence why you talking about resource limited setting? Why not you say simple ‘Ethiopia’. The title mislead readers.

Response: Studies carried out in settings with limited resources were thoroughly reviewed. Unfortunately, no studies have been conducted in environments other than Ethiopia that meet the inclusion criteria. Until articles are available, those initial research carried out in Ethiopia were thought to nominate the poor resource situation of low-income countries. Additionally, it is believed to be employed as an evidence-based practice in low resource settings to intervene with congenital abnormalities.

Discussion

The first and second paragraph is not related to discussion. The author should have focused on the summary finding in the first paragraph.

Response: Corrected accordingly

“Selection and characterization of included studies” what selection have to do in the result section?

Response: Title of the subsection was corrected. 

“Possible risk factors of Congenital anomalies in low resource setting” how authors identified risk factors using cross-sectional studies?

Response: The goal was to use "possible risk factors" rather than "risk factors," which might be used in the same way that we say "associated factors". 

Authors should link their finding to previous studies and provide the possible reason behind for the complement and disagreement of the findings. For instance how maternal illness related to tubal defect and what is the reason behind? How chat chewing related to tubal defect? Have you included studies from eastern part of the country where chat chewing is relatively high? What is the pathophysiology behind? Why https://www.ncbi.nlm.nih.gov/pmc/articles/PMC7672758/ not included? Which specific maternal illness is associated with tubal defects? How? Why?

Response: We apologize for the error in omitting such a significant paper from the study. However, we have now cited it. Since the primary studies that were included stated "maternal illness during pregnancy" as a variable, without identifying the type of illness; because of this we simply generalized it and assumed that any illness should be monitored cautiously to avoid congenital abnormalities. We advocate experimental research because the primary papers pooled in this analysis were quantitative observational studies that were unable to address some of the concerns outlined above, particularly the "how" ones. 

What are the implications of your findings?

Recommendations are not usually listed as you did, instead summarize in one or two sentences based on your conclusion.

Response: the recommendation part was revised. 

Additional files

Where is searching techniques for all databases?

Where is quality assessment grading using Newcastle-Ottawa Scale?

Response: provided as a supporting files. 

Thank you!

---

## [Editor Report · Decision Letter 1]

7 Sep 2023

Structural congenital anomalies in resource limited setting, 2023: A systematic review and meta-analysis

PONE-D-23-05213R1

Dear Dr. Geda Yohannes,

We’re pleased to inform you that your manuscript has been judged scientifically suitable for publication and will be formally accepted for publication once it meets all outstanding technical requirements.

Kind regards,

Abay Woday Tadesse

Academic Editor

PLOS ONE

Additional Editor Comments (optional):

The authors should improve the language of the manuscript, otherwise it is suitable for publication with current shape.
---

## [Editor Report · Acceptance letter]

6 Oct 2023

PONE-D-23-05213R1 

Structural congenital anomalies in resource limited setting, 2023: A systematic review and meta-analysis 

Dear Dr. Geda:

I'm pleased to inform you that your manuscript has been deemed suitable for publication in PLOS ONE. Congratulations! Your manuscript is now with our production department. 

Kind regards, 

on behalf of

Mr. Abay Woday Tadesse 

Academic Editor

PLOS ONE